# Surgical Options for Peritoneal Surface Metastases from Digestive Malignancies—A Comprehensive Review

**DOI:** 10.3390/medicina59020255

**Published:** 2023-01-28

**Authors:** Mihai Adrian Eftimie, Gheorghe Potlog, Sorin Tiberiu Alexandrescu

**Affiliations:** 1Department of General Surgery, Fundeni Clinical Institute, 022328 Bucharest, Romania; 2Department of Surgery, Faculty of Medicine, Carol Davila University of Medicine and Pharmacy, 050474 Bucharest, Romania

**Keywords:** peritoneal surface metastases, digestive cancers, colorectal cancer, gastric cancer, appendix cancer, cytoreductive surgery (CRS), hyperthermic intraperitoneal chemotherapy (HIPEC), pressurized intraperitoneal aerosolized chemotherapy (PIPAC)

## Abstract

The peritoneum is a common site for the dissemination of digestive malignancies, particularly gastric, colorectal, appendix, or pancreatic cancer. Other tumors such as cholangiocarcinomas, digestive neuroendocrine tumors, or gastrointestinal stromal tumors (GIST) may also associate with peritoneal surface metastases (PSM). Peritoneal dissemination is proven to worsen the prognosis of these patients. Cytoreductive surgery (CRS), along with systemic chemotherapy, have been shown to constitute a survival benefit in selected patients with PSM. Furthermore, the association of CRS with hyperthermic intraperitoneal chemotherapy (HIPEC) seems to significantly improve the prognosis of patients with certain types of digestive malignancies associated with PSM. However, the benefit of CRS with HIPEC is still controversial, especially due to the significant morbidity associated with this procedure. According to the results of the PRODIGE 7 trial, CRS for PSM from colorectal cancer (CRC) achieved overall survival (OS) rates higher than 40 months, but the addition of oxaliplatin-based HIPEC failed to improve the long-term outcomes. Furthermore, the PROPHYLOCHIP and COLOPEC trials failed to demonstrate the effectiveness of oxaliplatin-based HIPEC for preventing peritoneal metastases development in high-risk patients operated for CRC. In this review, we discuss the limitations of these studies and the reasons why these results are not sufficient to refute this technique, until future well-designed trials evaluate the impact of different HIPEC regimens. In contrast, in pseudomyxoma peritonei, CRS plus HIPEC represents the gold standard therapy, which is able to achieve 10-year OS rates ranging between 70 and 80%. For patients with PSM from gastric carcinoma, CRS plus HIPEC achieved median OS rates higher than 40 months after complete cytoreduction in patients with a peritoneal cancer index (PCI) ≤6. However, the data have not yet been validated in randomized clinical trials. In this review, we discuss the controversies regarding the most efficient drugs that should be used for HIPEC and the duration of the procedure. We also discuss the current evidence and controversies related to the benefit of CRS (and HIPEC) in patients with PSM from other digestive malignancies. Although it is a palliative treatment, pressurized intraperitoneal aerosolized chemotherapy (PIPAC) significantly increases OS in patients with unresectable PSM from gastric cancer and represents a promising approach for patients with PSM from other digestive cancers.

## 1. Introduction

Although most guidelines recommend only palliative oncologic therapy for patients with peritoneal surface metastases (PSM) of digestive origin, the responsiveness of PSM to systemic therapy is significantly lower compared to other metastatic sites [1,2]. Peritoneal implants are believed to be the consequences of primary tumor cell detachment or dissemination during surgical procedure [3].

As a consequence of the low survival rates achieved by palliative systemic therapy, there has been increased interest in the complete surgical removal of peritoneal deposits. The first surgical resection of PSM was performed for ovarian cancer. Cytoreductive surgery (CRS), first performed in the 1980s, represents the complete (or near-complete) removal of macroscopic disease. CRS, accompanied by hyperthermic intraperitoneal chemotherapy (HIPEC), has emerged as an aggressive and efficient loco-regional therapy. In the 1990s, Sugarbaker described the surgical technique for peritonectomy and associated visceral resections [4]. During the same period, various investigators developed drug regimens and methods for HIPEC according to the primary tumor site [5,6,7]. For selected patients, this approach offered improved survival rates and even better quality of life. Other forms of intraperitoneal chemotherapy such as early postoperative intraperitoneal chemotherapy (EPIC—on days 1–5) and sequential intraperitoneal chemotherapy (SIPC) are less commonly used.

The most important prognostic factors related to the overall survival (OS) of patients treated by CRS +/− HIPEC are Sugarbaker’s peritoneal cancer index (PCI), which quantifies the extent of the disease, and the completeness of the cytoreduction score (CC score), which evaluates the wholeness of the CRS. The impact of these parameters on OS rates depends on the site of the primary tumor and its histological type [8].

The completeness of the cytoreductive procedure has a direct impact on the survival of patients with PSM in most malignancies. Although the goal of CRS should be the achievement of a CC-0 score (no macroscopic residual tissue), at least in ovarian cancer, even a CC-1 score (persistent nodules less than 2.5 mm in largest diameter) seems to be associated with improved OS [9]. In order to achieve a CC-0/CC-1 score, a dedicated team involving surgeons, anesthesiologists, medical oncologists, and radiologists should evaluate the patient and subsequently perform the procedure, preferably in a high volume-center [10]. Preoperative chemotherapy seems to play an important role in the selection of patients who could really benefit from this aggressive procedure (CRS with or without HIPEC).

Even with optimal CRS, the majority of recurrences that occur are located intraperitoneally [9].

For patients with unresectable PSM, chemotherapy remains the gold standard treatment, even if its impact on survival is limited, ranging from 16.6 months for recurrent platinum-resistant ovarian cancer [11] to 16.3 months for colorectal carcinoma (CRC) [12], 10.7 months for gastric cancer [13], and less than 12 months for peritoneal mesothelioma [14]. For such patients, pressurized intraperitoneal aerosol chemotherapy (PIPAC) has been developed as a safe and well-tolerated palliative procedure that enhances the effect of chemotherapy (because of the physical properties of aerosol and pressure) and improves their OS [15].

With the increase in experience and the development of high-volume centers, the morbidity and mortality rates associated with these procedures has decreased, becoming similar to the respective rates of other major gastrointestinal surgeries [16]. A study conducted by Constance Houlze-Laroye [17] on 5562 patients, published in 2021, revealed that more than half of the postoperative deaths following CRS and HIPEC procedures were preventable.

Although the issue of CRS +/− HIPEC for PSM from specific malignancies has been addressed in other recent papers, there is a paucity of reviews that have presented, together, the latest evidence regarding the surgical options for all digestive carcinomas with peritoneal metastases. This paper aims to advance the field by informing current practice and by prompting clinicians to act and broaden the use of an aggressive surgical approach in patients with PSM from digestive carcinomas. Given the current evidence, concerted efforts should be made by general practitioners, gastroenterologists, oncologists, and surgeons to promote CRS with or without HIPEC in order to prolong the life-expectancy of these patients.

In this comprehensive review, the surgical approach of PSM is reported separately according to the primary site and/or histology. For each type of digestive malignancy, the patient selection protocols, specific approaches to CRS, HIPEC methodology and drug regimens, proper sequencing with other treatments, patient follow-up, and protocols used for recurrence are described and discussed. Furthermore, the paper reflects the most recent evidence regarding the prophylactic use of HIPEC in patients with colorectal or gastric carcinoma at high risk for developing PSM. We review the data critically, taking into account the limitations of the studies, and suggest future directions of research.

## 2. Paper Selection

We searched the PubMed database using the following terms: (((((((((((((peritoneal surface metastasis[Text Word]) OR (carcinomatosis[Text Word])) AND (colorectal cancer[Text Word])) OR (gastric carcinoma[Text Word])) OR (digestive malignancies[Text Word])) OR (biliary tract carcinoma[Text Word])) OR (pancreatic carcinoma[Text Word])) OR (gastrointestinal stromal tumors[Text Word])) OR (neuroendocrine tumors[Text Word])) OR (small bowel carcinoma[Text Word])) AND (cytoreductive surgery[Text Word])) OR (HIPEC[Text Word])) NOT (ovarian cancer[Text Word])) NOT (mesothelioma[Text Word]). The filters applied were: Clinical Trial, Meta-Analysis, Randomized Controlled Trial, Review, Systematic Review, from 1 January 2001 to 1 June 2022. The search generated 538 results. The abstracts of these results were evaluated by two authors (M.A.E. and G.P.), the relevant papers were extracted independently, and their full-text versions were assessed. Consensus for the relevance of a study was carried out by the third author (S.T.A.). We also evaluated the references of the relevant papers that were evaluated in order to identify additional articles that were not found during the initial search. Due to the heterogeneity of the studies, we report the results as a narrative review.

## 3. Surgical Options for PSM from Colorectal Carcinoma (CRC)

### 3.1. Epidemiology

CRC is the third most common type of cancer and generates the second most frequent cancer-related mortality globally. When diagnosed at an early stage, 70–80% of patients will benefit from a curative-intent surgical procedure, resulting in a 5-year survival rate of 72–93% for stages I–II [3].

For CRC, synchronous PSM is encountered in 6–7% of patients and almost half of them have peritoneal-only metastases [18]. Furthermore, the risk for metachronous PSM can be as high as 6% [19]. The literature reveals that an advanced T stage, the presence of positive lymph nodes, synchronous ovarian metastases, a poor differentiation of the primary tumor, a colonic versus a rectal origin, the R1/R2 resection of the primary tumor, the histologic type of mucinous or signet-ring adenocarcinoma, the perforation or stenosis of the primary tumor, and younger age are the most frequently reported risk factors for the development of metachronous PSM [19,20,21].

### 3.2. Treatment Options

Patients with PSM of colorectal origin have classically been treated only with systemic palliative oncologic therapy, and sometimes palliative surgery [3]. In patients who receive only palliative treatment, colorectal PSM is associated with a worse prognosis compared to non-peritoneal metastases (16.3 months for PSM vs. 19.1 months for liver-only metastases and 24.6 months for lung-only metastases [12,22].

In 2003, Verwaal et al. [23] published a Dutch phase 3 controlled trial comparing the OS rates achieved by CRS plus HIPEC vs. palliative surgery plus systemic chemotherapy in patients operated for bowel obstruction. They showed that the OS rates achieved by CRS plus HIPEC were significantly superior to those observed in patients treated with palliative surgery. Later on, many clinical protocols of CRS and HIPEC were evaluated in different high-volume centers to treat the patients with colorectal PSM. Thus, Elias D., Koga S., Quenet S. et al. [7,22,24,25] reported promising results for CRS and HIPEC when a macroscopically complete resection is performed (CC-0), with an average median OS of 40 months. In 2013, Goere D et al. [26] stated that, in specialized centers, CRS and HIPEC could even achieve a cure in one sixth of the patients who underwent a CC-0 resection, reporting 5-year disease free survival (DFS) rates of 16% in such patients. However, because all of these studies were retrospective, no definitive conclusions could be drawn, and most guidelines continued to recommend only palliative oncologic therapy in patients with PSM of colorectal origin, irrespective of the extent of peritoneal involvement.

To overcome this drawback, between February 2008 and January 2014, 265 patients were randomly assigned to CRS and HIPEC (133 patients) or to CRS alone (132 patients) in a randomized, open-label, phase 3 trial performed at 17 cancer centers in France (PRODIGE 7 trial). All patients were confirmed with CRC and PSM, had a PCI ≤ 25, a WHO performance status of 0 or 1, normal liver function, proper hematological function, and were eligible to receive chemotherapy for 6 months [27]. Any previous treatments were permitted, a 4-week wash-out period was indicated, and the main exclusion criteria were extraperitoneal metastases, previous HIPEC treatment, and grade 3 or worse peripheral neuropathy. For patients enrolled in the CRS plus HIPEC arm, the HIPEC technique was performed either in a closed or open abdomen manner, according to each center’s approach. Systemic chemotherapy (400 mg/m^2^ fluorouracil and 20 mg/m^2^ folinic acid) was administered intravenously 20 min before HIPEC (bidirectional chemotherapy protocol) and intraperitoneal chemotherapy consisted of oxaliplatin at a dose of 460 mg/m^2^ (for the open technique) or 360 mg/m^2^ (for the closed abdomen technique). Oxaliplatin was delivered intraperitoneally in 2 L/m^2^ of dextrose, heated at 43 °C, for 30 min. The follow-up was conducted one month after surgery, every 3 months for the first 3 years and every 6 months up to 5 years. The median OS was 41.7 months in the CRS plus HIPEC group and 41.2 months in the CRS alone group (*p* = 0.99). Although PRODIGE 7 did not reveal a survival benefit for the addition of HIPEC to CRS, this trial reported unexpectedly high OS rates in patients treated with CRS alone. These findings suggest that the completeness of CRS is the most important factor for survival in patients with PSM from CRCs, with similar observations already being reported by other authors in retrospective studies [23,24,28]. Furthermore, median relapse-free survival (RFS) between the two groups was not significantly different and 15% of patients in each group were considered cured at 5 years. According to the data of the PRODIGE 7 trial, CRS alone should be the cornerstone of therapeutic strategies with curative intent for colorectal peritoneal metastases [23], and the benefit of HIPEC is still debatable.

### 3.3. Prognostic Factors in Patients Treated with CRS +/− HIPEC

The only significant survival difference between the two study arms of the PRODIGE 7 trial was found in the subgroup of patients with a PCI between 11 and 15. In these patients, CRS and HIPEC were associated with significantly higher RFS rates than CRS alone, although the OS rates were similar among the two study arms. This might be the basis for further studies aiming to evaluate a potential survival benefit offered by CRS plus HIPEC vs. CRS alone in patients with more extensive PSM involvement.

Moreover, the cut-off value of the PCI associated with a significantly higher survival benefit after CRS + HIPEC has not been uniformly reported by different authors. Thus, Gustave Roussy’s group revealed that the maximum survival benefit of CRS plus HIPEC was achieved in patients with a PCI ≤ 10. [26] The Consensus Guidelines from The American Society of Peritoneal Surface Malignancies on standardizing the delivery of hyperthermic intraperitoneal chemotherapy (HIPEC) in CRC patients in the United States, published in 2014 [29], state that CRS is particularly effective in patients with a low-volume peritoneal disease, suggesting that a PCI ≤ 12 and no evidence of systemic disease are the main prognostic factors for better survival. Yan TD [30] stated that patients with a PCI ≤ 13 had a better life expectancy. Authors such as Da Silva and Sugarbaker [31] set the limit of the PCI at 20. This value is also supported by data from Cavaliere et al. [32] and Van Sweringen et al. [33], whose data indicated that a PCI > 20 is associated with decreased survival rates, hence, they concluded that such patients should not be seen as candidates for CRS +/− HIPEC.

In and by itself, the PCI cannot predict unresectability for certain tumor locations [34]. Thus, some studies have suggested that the number of regions affected by PSM of colorectal origin and invasion of the small bowel in more than two different parts are independent prognostic factors for both unresectability and shorter survival [35,36]. A paper published by Elias D. et al. [37] in 2014 also revealed that the involvement of the lower ileum and a high PCI were negative prognostic factors for the efficacy of the multimodality treatment, while Verwaal et al. [23] demonstrated a clear decrease in the survival rates in patients with PSM involving six or more regions (*p* < 0.0001).

Alongside the CC score, the PCI, and the number of regions with PSM, other studies have suggested additional prognostic factors that were independently associated with OS and/or DFS in patients who underwent CRS +/− HIPEC for PSM of colorectal origin. However, the impact of these additional prognostic factors is still controversial, since most of the data are from relatively small retrospective studies. For example, Tonello M. et al. [38] found that operated patients with PSM of rectal origin had a worse prognosis than those with PSM of colonic origin. Hence, they proposed a more restrictive use of CRS and HIPEC in patients with PSM of rectal origin. The impact of the location of the primary tumor on OS in patients with PSM of colorectal origin was also assessed by Peron et al. [39] in a prospective study that included 796 patients undergoing complete CRS (CC-0) between January 2004 and January 2017 in 14 institutions from France (the BIG-RENAPE database) and two institutions from Canada. They revealed that the primary site had no impact on the long-term outcomes of patients with PSM undergoing a complete CRS. No impact on OS and DFS was encountered across all subgroups of patients. This study also found no impact of RAS and BRAF mutations on the outcomes after complete CRS. This evidence suggests that the side of the primary tumor should not represent an exclusion criterion for patients with PSM from colorectal origin that are amenable to CRS (with or without HIPEC). Similar results were reported by Massalou et al. [40], who found that the location of the primary tumor location as well as RAS and BRAF status had no significant impact on the OS or DFS. In their study, the only pathologic/molecular factors associated with worse OS after CRS + HIPEC were the signet ring and mucinous type of carcinoma, while the presence of microsatellite sequence stability (MSS) was associated with lower DFS rates. This study also found that BMI > 25 was associated with significantly lower OS and DFS rates.

### 3.4. Morbidity and Mortality after CRS +/− HIPEC

Higher BMI is also correlated with increased postoperative morbidity and mortality rates in colorectal procedures including CRS with or without HIPEC [41,42]. Regarding the 30-day mortality rates after CRS with/without HIPEC, most studies reported an average value of 2% [16,24,25,27]. In the PRODIGE 7 trial, there was no statistically significant difference (*p* = 0.083) concerning the frequency of grade 3 or worse adverse events at 30 days between the CRS alone group (32%) and the CRS + HIPEC group (42%) [23]. Similarly, Foster et al. [43] used the data from the American College of Surgeons National Surgical Quality Improvement Project database and found that CRS and HIPEC were associated with perioperative and 30-day postoperative morbidity and mortality rates similar to those of other oncological surgical procedures. However, the PRODIGE 7 trial showed a significantly increased 60-day rate of grade 3 or worse complication in the CRS plus oxaliplatin-based HIPEC group vs. the CRS alone group (26% vs. 15%, respectively; *p* = 0.035) [23]. This indicates that patients in the CRS + HIPEC group have a longer period of risk for developing complications, leading to a prolonged time to resumption of postoperative systemic chemotherapy. The lack of survival benefit and the significantly higher rate of grade 3 or worse adverse events at 60 days in the CRS + HIPEC group (vs. CRS alone group) seem to be reasonable arguments to refute the use of prophylactic HIPEC in patients with non-metastatic CRC at risk of developing PSM [27].

### 3.5. HIPEC Protocol

Although the PRODIGE 7 randomized controlled trial did not find any survival benefit from the association of HIPEC to CRS in patients with PSM of colorectal origin, its results were critically appraised by many authors including Paul H. Sugarbaker [44]. The major criticism of the PRODIGE 7 trial was related to the HIPEC protocol, concerning both the dose and the duration of chemotherapy.

In the PRODIGE 7 trial, the oxaliplatin-based HIPEC regimen was limited to 30 min. Kirstein MN [45] and Lemoine L [46] demonstrated that the response to local oxaliplatin was related to the duration of exposure. Furthermore, Levine EA et al. [47] used a HIPEC regimen lasting 120 min in their study, while Van Driel WJ [48] opted for a 90 min cisplatin-based HIPEC protocol for the treatment of ovarian cancer. Both reported increased overall survival rates with prolonged duration of HIPEC.

Regarding the cytotoxic agent used for HIPEC in the PRODIGE 7 trial, several concerns have been raised, because no standard regimen exists thus far. Hence, to increase the efficacy of intraoperative chemotherapy, many protocols have been put in place [49]. For example, the intensification of the HIPEC regimen with irinotecan has been explored in a previous study, but could not be associated with any survival benefit [25]. Furthermore, a cisplatin-based HIPEC protocol was associated with inferior long-term outcomes compared to an oxaliplatin-based regimen in an Italian multicentric study conducted by Cavaliere [32]. Thus, the most frequently used HIPEC regimens are based on oxaliplatin or mitomycin C. A Dutch series reported by Hompes et al. [50] as well as a large American retrospective study conducted by Prada-Villaverde [51] suggested no significant differences in the OS rates between the oxaliplatin-based and mytomicin C-based protocols. However, a single-center Australian study reported superior OS rates achieved by an oxaliplatin-based HIPEC regimen compared to the mytomicin C-based protocol [52]. The major criticism regarding the use of the oxaliplatin-based HIPEC protocol in the PRODIGE 7 trial is related to the extensive use of oxaliplatin in these patients before HIPEC. Previous studies [53,54] have suggested that the patients hard-treated with oxaliplatin could develop oxaliplatin resistance, resulting in decreased rates of response to a further oxaliplatin-based regimen. In the PRODIGE 7 trial, extensive oxaliplatin treatment before surgery might induce misleading results in the arm of patients treated with CRS + HIPEC, raising the question of whether a mitomycin-C based HIPEC regimen or an oxaliplatin-based HIPEC regimen prolonged to 120 min would be associated with higher survival rates in this arm.

### 3.6. Recurrent PSM

Despite the aggressive approach and curative intention, between 70% and 80% of patients with colorectal PSM treated by CRS (alone or combined with HIPEC) will develop recurrent disease [7,55]. This has led to the idea of iterative CRS and even HIPEC procedures. Several studies have suggested that in high-volume centers, the morbidity and mortality associated with these procedures are similar to those of the initial intervention [56]. This aggressive approach has led to a moderate increase in the median OS from 39 months to 42.9 months when compared to systemic treatment alone [3,56,57,58]. Although HIPEC has not been proven to be an independent risk factor for the development of postoperative complications [59], its benefit in the treatment of recurrent PSM from CRC needs further evaluation in prospective randomized controlled trials.

### 3.7. Prophylactic HIPEC in High-Risk Patients

Proactive strategies regarding high-risk patients with CRCs are still a matter of debate and no strong evidence supports their superiority versus proper surveillance. Authors such as Dominique Elias [60,61] and Serrano Del Moral [62] suggest that second-look surgery in conjunction with imagistic investigations, colonoscopies, and CEA level surveillance for high-risk patients can offer the early detection of PSM and precocious aggressive treatment. The promising results associated with prophylactic resection of target organs during the primary surgery (omentectomy, hepatic round ligament resection, appendicectomy, adnexectomy) [63] or prophylactic HIPEC administration at the time of the primary procedure for advanced tumors without PSM [64,65,66] represent the basis for some phase III randomized clinical trials evaluating the usefulness of such approaches (e.g., the ProphyloCHIP trial and COLOPEC trial). The PROPHYLOCHIP-PRODIGE 15 trial [67] evaluated the impact of second-look surgery and HIPEC vs. follow-up on 3-yr DFS of patients with resected CRC and high-risk of developing PSM (perforated primary tumor/peritoneal or ovarian metastases radically resected concomitant with CRC). The authors did not find a significant difference in the 3-yr DFS rates (44% vs. 53%, respectively; *p* value = 0.82). On the other hand, the COLOPEC trial [68] assessed the role of adjuvant HIPEC in preventing the occurrence of peritoneal metastases in patients with resected T4/perforated primary tumor, who received adjuvant systemic chemotherapy. There was no statistically significant difference in 18-months peritoneal DFS rates between patients treated with adjuvant systemic therapy only (76.2%) and those treated with adjuvant HIPEC and systemic therapy (80.9%; *p* value = 0.28). However, in both of these studies as well as in the PRODIGE 7 trial, the HIPEC protocol consisted in the administration of oxaliplatin only for 30 min. Although these trials have generated skepticism toward the usefulness of HIPEC, these results could be challenged by ongoing/future trials evaluating the different protocols of HIPEC. Until new HIPEC protocols are tested in well-designed comparative trials, this procedure should not be considered as an ineffective method [69].

Take home message: Complete CRS represents the cornerstone therapy in patients with PSM from colorectal carcinoma and a low PCI. The addition of HIPEC to complete CRS in such patients seems to have a limited benefit and this approach should be restricted to patients with a PCI > 10, operated in specialized centers, and preferably in the context of controlled trials. The current results cannot support the routine use of prophylactic HIPEC in patients operated for colorectal carcinoma with a high-risk for the development of PSM (T4/perforated primary).

## 4. Surgical Options for PSM from Gastric Carcinoma

### 4.1. Epidemiology

Gastric cancer is the third leading cause of cancer deaths worldwide and has the fifth highest incidence among solid cancers in adults. PSM from gastric adenocarcinoma is found in 17% of newly-diagnosed patients and is associated with a poor prognosis. Advanced stages such as stage III gastric adenocarcinomas can be associated in up to 40% of cases with PSM [70,71].

### 4.2. Treatment Modalities

According to the NCCN guidelines, the treatment options for patients with PSM from gastric carcinoma include palliative systemic therapy, supportive treatment, and surgery for complications [72]. Despite recent advances in oncologic therapy (e.g., trastuzumab in patients with HER-2/neu gene amplification, check-point inhibitors), the median overall survival of patients with PSM of gastric origin ranges from 8 to 10 months [73].

Due to the dismal prognosis associated with the current oncologic therapy, aggressive surgical approaches have been developed in the last two decades including CRS in combination with HIPEC as a potentially curative-intent therapy and PIPAC as a palliative therapy able to prolong survival in patients with unresectable PSM or high PCI. The CRS + HIPEC approach should be combined with neoadjuvant and adjuvant systemic therapy in order to better select patients and increase the DFS and OS rates.

The Japanese group of Yokemura suggested for the first time, in the 1990s, the feasibility and efficacy of CRS plus HIPEC for the treatment of PSM from gastric adenocarcinoma [74,75]. These findings were further supported by a European series reported by Glehen et al. [76]. In 2010, a large retrospective multicentric study from France (159 patients) added more evidence to the concept of CRS and HIPEC for the treatment of PSM of gastric origin. This study showed that complete CRS (CC-0 score) [77,78] was an independent predictor of prolonged OS. Thus, the median OS in patients who underwent CC-0 was 15 months, significantly higher than those achieved in the entire group of patients, irrespective of the completeness of cytoreduction (9.2 months). Furthermore, when complete CRS (CC-0) was achieved, the 5-year OS rate was 23%. In a large retrospective study with propensity score matching analysis, Glehen et al. [79] showed that incomplete CRS (CC-1) is associated with a 5-year OS rate of 6.2%, significantly lower than the 24.8% 5-year OS achieved by complete CRS (CC-0). Similarly, Coccolini et al. [80] found significantly higher 1- and 3-year OS rates in patients who underwent CC-0, compared to those achieved by CC-1. Both studies revealed that complete CRS (CC-0) was correlated with the initial tumor burden expressed by the value of the PCI. Similarly, Yonemura et al. evaluated 95 patients and found that CC-0 had been achieved in 91% of patients with a PCI ≤ 6, but only in 42% of the patients with a PCI ≥ 7. Furthermore, the OS rates were significantly higher in patients with PCI ≤ 6 compared to patients with a PCI > 6 [78]. The study of Cambay et al. [81] reported similar results. A more recent multicentric study from Italy, which included 91 patients with gastric carcinoma and synchronous PSM, reported median OS rates higher than 40 months in patients with a PCI ≤ 6 as well as in those who underwent complete cytoreduction [82]. Thus, the median OS after CC-0 was significantly higher compared to the OS of patients with incomplete resection (40.7 vs. 10.7 months, respectively; *p* value = 0.003). Moreover, in patients with a PCI > 6. the median OS was significantly lower than in patients with a PCI ≤ 6 (13.4 vs. 44.3 months, respectively; *p* value = 0.005) and the mortality was almost double [82].

### 4.3. Prognostic Factors in Patients Treated with CRS +/− HIPEC

Multiple retrospective studies from Italy, Spain, Germany, and Central-Eastern European countries have supported the observation that complete CRS (CC-0) and a low PCI are the main independent prognostic factors associated with better prognosis in this type of approach [82,83,84,85].

However, there is no universally-accepted cut-off value of the PCI to select patients with PSM of gastric origin for CRS plus HIPEC. Although most centers recommend such an aggressive surgical approach in patients with a PCI ≤ 6, some high-volume centers suggest that even in patients with a PCI between 7 and 12. there is a survival benefit from CRS + HIPEC [76,86,87].

Other negative prognostic factors such as signet ring cell histology, presence of lymph node metastasis, and lack of tumor regression after preoperative chemotherapy were revealed by these studies. A recent multicenter study by the “Italian Peritoneal Surface Malignancies Oncoteam—S.I.C.O.” proved the beneficial effect of neoadjuvant chemotherapy on the long-term outcomes of patients eligible for CRS and HIPEC [82]. The same group highlighted a significant negative prognostic effect determined by positive peritoneal cytology [82,88].

However, the prognostic impact of signet ring cell histology on the long-term outcomes of patients treated by CRS plus HIPEC for PSM of gastric carcinoma is still debatable. In 2014, Konigsrainer et al. [89] hypothesized that for patients with PSM from gastric cancer with signet ring cell histology, CRS + HIPEC should not be considered due to the high recurrence rates. However, these authors did not support this hypothesis with evidence derived from a specific study. In 2019, Solomon et al. [90] revealed the negative impact of the signet ring cell histologic subtype on the OS of patients treated with CRS + HIPEC for PSM of various origins, but surprisingly, in PSM from gastric cancer, the OS was not significantly different between patients with signet ring cell pathology and those with other pathologic subtypes (*p* = 0.245). Similarly, a Spanish study published in 2018 found that the only prognostic factor that was independently associated with worse OS after CRS + HIPEC for PSM of gastric origin was perineural invasion (HR = 18.886, 95% CI: 1.104–323.123; *p* = 0.043), while the signet ring cell subtype did not significantly influence the OS [83].

Because most of these studies were retrospective and had a small sample-size, definitive conclusions on the real benefit of CRS + HIPEC for PSM of gastric origin cannot be drawn. The most reliable conclusions on this topic should probably be derived from the results of the CYTO-CHIP study, an observational study that included 277 patients from 19 French centers [79]. This is the largest study published thus far to assess the comparative results of CRS vs. CRS + HIPEC in patients with PSM from gastric carcinoma. Similar to the previously-mentioned studies, complete CRS (CC-0) was associated with significantly higher 5-year OS rates compared to CC-1 (24.8% vs. 6.2%, respectively; *p* < 0.05), and lower PCI was confirmed as an independent prognostic for better OS. The most important findings of this study are the significantly higher OS and DFS rates achieved by CRS + HIPEC compared to CRS alone, without significant increase in major morbidity and 90-day mortality. These results support the performance of CRS + HIPEC when CC-0 can be achieved in patients with limited PSM of gastric origin [79]. Furthermore, the study suggested that CRS + HIPEC performed in specialized centers was associated with morbidity rates similar to those reported after other aggressive surgical procedures [91].

The results of an ongoing phase III randomized controlled trial (PERISCOPE II), which compares the CRS + HIPEC vs. palliative systemic therapy in patients with gastric cancer and limited peritoneal dissemination or positive peritoneal cytology, will be able to improve the current knowledge on this topic, assuming or rejecting the current hypothesis about the usefulness of CRS + HIPEC [92].

### 4.4. Prophylactic HIPEC in High-Risk Patients

Gastric cancer is associated with a high risk for developing PSM. Around 50% of patients with potentially curable advanced gastric cancer die from recurrence in the peritoneum [93]. A total of 15 to 50% of patients with serosal involvement present peritoneal dissemination at the time of the initial surgical exploration [94].

A study by Seyfreid et al. [95] on 1108 patients that were treated for gastric cancer with radical D2 gastrectomy revealed a 50% recurrence rate. Out of these patients, 15.5% developed metachronous PSM after a median time of 17.7 months. The major risk factors for PSM were found to be serosal involvement, the extent of nodal metastasis, and tumor pathology—signet ring cell and undifferentiated carcinoma.

Furthermore, the Japanese General Rules of Gastric Cancer Treatment divide PSM into two categories with the same prognosis [96,97]: (1) P0/Cy1—positive peritoneal wash cytology; (2) P1—macroscopic PSM.

Due to the high-risk of developing PSM in patients with such risk factors, some authors hypothesized that adjuvant HIPEC might be associated with decreased rates of recurrence and improved survival.

Jingxu Sun et al. [98] performed a meta-analysis on 280 studies analyzing the impact of adjuvant HIPEC on the prognosis of patients with serosal involvement from gastric cancer and found that HIPEC improved the long-term outcomes of these patients, with acceptable morbidity and mortality rates. Similarly, a 2019 study on 80 locally advanced gastric tumor patients (T stage ≥ 3) with no signs of PSM or systemic disease, conducted by Maneesh Kumarsing Beeharry [99], proved that the combination of radical gastrectomy with HIPEC has been associated with acceptable complication rates and improved the OS rates.

To evaluate these results in a European cohort of patients, a randomized multicenter phase III trial (GASTRICHIP) was initiated. This study aimed to evaluate the effects of HIPEC with oxaliplatin on patients with gastric cancer involving the serosa and/or lymph nodes and/or with positive peritoneal cytology, treated with perioperative systemic chemotherapy and D1-D2 curative gastrectomy [100].

Take home message: Current evidence supports the performance of CRS + HIPEC in carefully selected patients with a PCI ≤ 6, when CC-0 can be achieved in high-volume centers. Prophylactic HIPEC in patients with gastric carcinoma at high-risk of PSM development should not be routinely recommended until the results of ongoing trials are made available.

## 5. Surgical Options for PSM from Pseudomyxoma Peritonei (PMP)

Pseudomyxoma peritonei (PMP) is a rare peritoneal malignancy, most commonly originating from a perforated epithelial tumor of the appendix, also known as “Jelly Belly” and is characterized by the bulky accumulation of gelatinous tumor deposits in the peritoneal cavity.

CRS and HIPEC represent the gold standard treatment for PMP. The main factors that influence a patient’s outcome are the histological type and the completeness of the cytoreduction. Thus, the peritoneal mucinous carcinomatosis (PMCA) histologic subtype is associated with significantly worse prognosis compared to the diffuse peritoneal adenomucinosis (DPAM) subtype or hybrid tumors [101]. For appendicular PMP, complete CRS (CC-0) combined with HIPEC was associated with 5- and 10-year OS rates of 85% and 75%, respectively [101,102]. The most frequently used HIPEC regimens are based on oxaliplatin or mitomycin C. However, Chua et al. found that HIPEC was significantly associated with an improved rate of PFS, but it had no significant impact on the OS rates. Thus, even though HIPEC may improve disease control, optimal cytoreduction seems to be the strongest predictor of long-term survival [101].

Some intraoperative findings such as the involvement of the hepatic hilum [34,103], the infiltration of the anterior pancreatic surface [104,105], the ureteric obstruction, or the need for complete gastric resection [106] can impede the achievement of complete CRS. In such instances, although incomplete CRS is known to be associated with significantly decreased OS rates compared with complete CRS, patients with appendiceal PMP seem to benefit from CC-1 resections (remaining nodules smaller than 2.5 mm) and even debulking surgical procedures [102]. The concept of “maximum tumor debulking” (MTD) has been accepted as an alternative to CC-0/CC-1 resection, when complete CRS is not possible or in patients who are not fit for complex surgery [107]. MTD usually involves a greater omentectomy, lower abdominal peritonectomies. and an extended right hemicolectomy, usually associated in women with bilateral oophorectomy [102]. Several studies have shown that MTD plus HIPEC is feasible (achieving low morbidity and mortality rates) and is associated with acceptable OS rates (5-year OS ranging between 24% and 46% after CC-2 or CC-3 resection, compared to 80% after CC-1 resection) [101,108,109,110].

Absolute contraindications to CRS and HIPEC in patients with PMP are extensive small bowel serosa involvement (at least 1.5 m of small bowel must remain after surgery) [111,112] and mesenteric retraction and infiltration.

Take-home message: In patients with PMP, complete CRS (CC-0) or near-complete CRS (CC-1) associated with HIPEC represents the gold standard therapy. The concept of “maximum tumor debulking” has been accepted in PMP as an alternative to CC-0/CC-1 resection, when complete CRS is not possible, or in patients who are not fit for complex surgery.

## 6. Surgical Options for PSM from Pancreatic Adenocarcinoma

PSM originating from pancreatic cancer are generally considered incurable and the only treatment option is palliative treatment. PSM is found in approximately 40% of patients, but free intraperitoneal tumor cells are detected in an additional one third of the cases without macroscopic PSM [113,114].

There is a lack of evidence regarding the possible benefits of CRS and HIPEC in patients with PSM from pancreatic cancer. Tentes et al. performed complete CRS or near-complete CRS with HIPEC in seven cases of PSM from pancreatic tail adenocarcinomas and four patients survived for more than 12 months without evidence of recurrence. They suggest that CRS with HIPEC may be considered as a treatment option for highly selected patients with pancreatic cancer and peritoneal metastases [115].

In addition, there is a series of patients with prophylactic use of HIPEC after R0 resection of pancreatic cancer, without peritoneal metastasis. Survival results achieved by this approach are among the highest reported in patients treated with curative intent for pancreatic adenocarcinoma [116]. However, Larentzakis et al. concluded that more controlled studies are needed to justify the use of HIPEC as a prophylactic therapy in resectable pancreatic adenocarcinoma, while CRS and HIPEC for the treatment of PSM of pancreatic origin seems to be useless (and possibly unsafe) at this level of evidence [117].

Take-home message: In patients with pancreatic adenocarcinoma, current evidence cannot support either the performance of CRS +/− HIPEC in the case of PSM, or the prophylactic use of HIPEC in high-risk patients, outside the controlled clinical trials.

## 7. Surgical Options for PSM from Biliary Tract Carcinoma

PSM from biliary carcinoma is associated with poor outcomes. The treatment for the majority of cases does not imply a surgical gesture and consists of palliative chemotherapy. Amblard et al. compared the impact on survival of CRS and HIPEC (34 cases) with palliative chemotherapy for patients with PSM from biliary carcinoma (25 cases) [118]. The median PCI in the surgical group was 9 (3–26). Macroscopically complete resection could be achieved in 25 patients (73%). Median OS and 3-year OS rate were 21.4 months and 30% in the CRS plus HIPEC group and 9.3 months and 10%, respectively, in the chemotherapy group. The authors concluded that CRS plus HIPEC could be considered for selected patients with a good performance status, low burden of disease, and PSM amenable to complete CRS [1].

Take home message: Currently, surgery for PSM from biliary carcinoma is controversial and future prospective/randomized controlled trials are needed before recommending such an aggressive approach, even in selected patients.

## 8. Surgical Options for PSM from Gastrointestinal Stromal Tumors (GISTs)

Gastrointestinal stromal tumors (GIST) are the most common mesenchymal neoplasms of the gastrointestinal tract. Surgery is the most effective treatment for resectable primary GIST without metastasis. Approximately 15–47% of patients present with overt metastatic disease with the most common sites of metastases being the liver, peritoneum, and omentum [119].

Surgical treatment for patients with metastatic gastrointestinal stromal tumors remains controversial. Prior to the introduction of systemic treatment with imatinib, outcomes for metastatic GIST were poor, median survival ranging between 10 and 20 months with 5-year OS rates lower than 10% [120]. With the introduction of imatinib in 2002, patient outcomes improved, with an acceptable systemic toxicity [121]. However, imatinib is not a curative treatment and needs to be associated with cytoreductive surgery to achieve better long-term outcomes.

Some retrospective studies [122,123] have reported that tumor size is an important factor in imatinib resistance. An et al. [124] reviewed 249 advanced GIST patients (102 patients with metastatic disease and 147 with multifocal disease relapse) and compared the outcomes achieved by CRS (more than 75% of the initial tumor bulk removed) vs. no CRS, prior to imatinib treatment. They found that CRS was not associated with better long-term outcomes. Their data suggest that cytoreductive surgery prior to imatinib treatment has no benefits for the outcome of the patient.

Thus, for most patients with metastatic GIST, imatinib is the first treatment option. The role of CRS in patients with metastatic GIST with variable responses to imatinib is still debated. Several studies have concluded that patients with disease response to tyrosine kinase inhibitor (TKI) treatment benefit more from CRS (R0/R1) than those with disease progression on TKIs [120,125,126,127,128,129,130]. Similarly, a multicenter retrospective study from Spain compared the long-term outcomes observed in two cohorts of patients (treated with CRS or without surgery) who achieved partial response (PR) or stable disease (SD) after initial imatinib treatment. This study reported lower median OS in the imatinib only group (59.9 months) compared to the imatinib and CRS group (87.6 months) [131].

CRS (R0/R1) for patients who respond to TKI should be considered no earlier than 6 months after starting the initial systemic therapy (in order to evaluate if they have PR or SD), but not later than 2 years after TKI initiation. TKI treatment should be resumed postoperatively [125,132,133].

The benefits of CRS for imatinib-resistant metastatic GIST are controversial. Several studies have shown that patients who undergo surgery for the focal progressive disease have a limited benefit [126]. However, for imatinib-resistant patients, sunitinib as a second-line therapy seems to be the most appropriate treatment option. The surgical management of patients with progressive metastatic GIST receiving sunitinib is even more controversial, although Yeh et al. [134] and Raut et al. [135] suggest that surgery is feasible and safe for highly selected patients with metastatic GIST who are receiving sunitinib.

Take-home message: CRS should be considered in patients with metastatic GIST whose disease responds to imatinib, with the goal of performing R0/R1 resection. However, debulking/palliative surgery should be limited to patients with complications due to PSM from GISTs (such as hemorrhage, pain or intestinal obstruction) [136]. According to most authors, the role of HIPEC for the treatment of PSM from GIST is still difficult to determine [137,138,139].

## 9. Surgical Options for PSM from Gastroenteropancreatic Neuroendocrine Tumors (GEP-NETs)

The incidence of PSM in patients with GEP-NETs is approximately 20%. Most of these metastases originate from primary tumors located in the midgut [140], especially in the ileum, and are often associated with other metastatic sites such as liver metastases, mesenteric lymph nodes, lung and bone metastases [141,142]. Hepatic involvement and tumor grade are the most important prognostic factors [143].

Complete CRS is the best option for patients with metastatic GEP-NETs and appears to improve patient outcomes [144,145,146,147,148,149,150]. Therefore, primary tumor resection should be performed during CRS [145,151]. All PCI levels were considered suitable for surgery if resectable [146]. Multivisceral resections and peritonectomy could be part of CRS, and in most cases, are associated with liver metastasis resection and/or radiofrequency ablation and the radiologic chemo-embolization of liver metastases [146,149].

Some studies have shown that a 90% decrease in tumor volume after CRS is associated with the best OS rates [150,152]. Recently, the cytoreduction level has been lowered to 70% of the initial tumor burden, according to several studies that have demonstrated a significant survival benefit for this level of cytoreduction [148,153,154].

The role of HIPEC remains undetermined in patients with PSM from GEP-NETs and a randomized study to evaluate the impact of HIPEC should be initiated [137,147,155].

Take-home message: Complete resection of PSM from GEP-NETs is recommended whenever possible. Patients whose PSM cannot be completely resected seem to achieve a significant survival benefit with debulking surgery, if at least 70% of the tumor burden can be removed. The potential benefit of HIPEC is still unknown.

## 10. Surgical Options for PSM from Small Bowel Adenocarcinoma

Small bowel cancer is a rare malignancy comprising less than 5% of all digestive cancers. Adenocarcinoma is a frequent subtype, accounting for 37% of all small bowel cancers [156]. Although surgical resection of the primary tumor is the mainstay of treatment management for localized disease, the recurrence rates remain as high as 40% [157]. Furthermore, approximately one third of patients present with stage IV disease [158]. One of the most frequent sites of metastatic involvement in patients with small bowel adenocarcinoma (SBA) is the peritoneal surface, especially in tumors arising from the jejunum and ileum. Other common metastatic sites include liver, lymph nodes, and lungs [159,160,161,162].

The prognosis of metastatic SBA is poor, with a 5-year survival rate of 15–33% and a median OS ranging from 12 to 20 months [156,163,164,165]. Six comparative studies showed a higher median OS in patients who received chemotherapy (12–16 months) versus patients who did not receive chemotherapy (2–8 months) [160,166].

PSM from SBA represents a therapeutic challenge. Several studies showed that CRS and HIPEC improved the outcomes for selected patients with PSM from SBA, achieving a median OS of 31–32 months [159]. The goal of CRS should be the achievement of CC-0. Patients who received complete CRS (CC-0) had a median OS of 43 months, significantly higher than those achieved by CC-1, CC-2, or CC-3 [167].

The following significant prognostic variables associated with improved survival after CRS plus HIPEC were reported: resection of the primary tumor before CRS plus HIPEC, time interval shorter than 6 months between the detection of PSM and CRS plus HIPEC therapy, well-differentiated tumor, absence of lymph node metastasis, absence of extraperitoneal metastasis, normal value of CA 125 and CA 19-9, absence of ascites, a PCI ≤ 15, achievement of CC-0, absence of postoperative complications, and oxaliplatin-based regimen of HIPEC [160,162]. Oxaliplatin-based HIPEC showed a significant survival advantage over the mitomycin C-based HIPEC regimen [162].

Levine et al. suggest that earlier surgical intervention is likely to be more effective than those performed after extensive systemic chemotherapy [168,169]. The most frequently used regimens of systemic chemotherapy are FOLFIRI, FOLFOX, and CAPOX [167,170,171,172].

Take home message: Based on the available evidence, complete CRS (CC-0) plus HIPEC seems to be safe and more beneficial than systemic chemotherapy alone in selected patients with PSM from SBA. However, future larger studies are needed before routinely recommending this aggressive approach.

## 11. Pressurized Intraperitoneal Aerosolized Chemotherapy (PIPAC)

PIPAC, a palliative surgical technique designed to deliver chemotherapy (cisplatin, doxorubicin, oxaliplatin) into the peritoneum under pressure, has recently been added to the armamentarium of oncologists to address PSM in patients who are not eligible for CRS [173]. The first report of the successful application of PIPAC in three patients with PSM was published in 2014 [15], and since then, a small number of articles have described the effectiveness and safety of PIPAC for the treatment of PSM in patients with cancers of various origins, the most common being gastric cancer [174].

Systemic chemotherapy is the gold standard approach for unresectable PSM, even if its impact on survival is limited [175]. The expected median survival is estimated at 16.3 months for CRC [176] and 10.7 months for gastric cancer [13], while with PIPAC, the median survival in patients with PSM of gastric origin increases up to 15.4 months, according to the number of PIPAC procedures [177,178,179].

Alyami et al. [175] reported that complete CRS and HIPEC could be achieved after repeated PIPAC sessions in carefully selected patients with unresectable PSM at diagnosis. In their cohort, the median PCI was 16, all patients underwent systemic chemotherapy between PIPAC sessions, the median consecutive PIPAC procedure was 3 (1–8), and 14.4% of patients were eligible for a secondary CRS and HIPEC after being considered unresectable prior to PIPAC.

A study published by Girshally et al. [180] suggested that neoadjuvant PIPAC is feasible and can be considered before CRS/HIPEC in a select group of patients with PSM of gastric origin and small bowel involvement, in order to reduce the extent of CRS. In their cohort, 12 out of 21 patients had a low PCI (mean 5.8 ± 5.6) and the remaining nine patients had advanced peritoneal involvement (mean PCI 14.3 ± 5.3) at the initial laparoscopy. Repeated PIPAC (3–4 cycles per patient) led to radiological tumor regression in seven out of nine patients, while major histological regression was achieved in eight out of nine patients, allowing for the subsequent performance of CRS + HIPEC.

The PIPAC procedure for PSM from non-gastric cancers is controversial and there is a paucity of data related to the role of PIPAC in PSM of non-gastric origin. Di Giorgio et al. reported that PIPAC with cisplatin, doxorubicin, or oxaliplatin is safe and has antitumor activity against peritoneal metastases of pancreatic and biliary tract origin [181].

Take home message: PIPAC seems to be a valuable palliative approach in patients with unresectable PSM of gastric origin, and is able to significantly prolong the survival of these patients. For PSM from other digestive malignancies, PIPAC requires more prospective controlled trials to better define its role in the palliative treatment of such patients.

## 12. Conclusions

The aggressive surgical approach of PSM from digestive malignancies, consisting of CRS with or without HIPEC, has gained wider acceptance during the last decade, especially in patients with CRC or gastric carcinoma. This is the consequence of the evidence offered by high-quality randomized clinical trials and meta-analysis that revealed that CRS is the cornerstone therapy in patients with PSM from CRC, although the oxaliplatin-based HIPEC regimen failed to further improve the survival of these patients. Supplementary well-designed randomized trials testing new HIPEC regimens are needed before refuting this therapy. Similarly, in patients with PSM from gastric carcinoma, future randomized controlled trials are needed to confirm the favorable outcomes achieved by CRS with HIPEC in large retrospective studies and meta-analysis. While in PMP the role of CRS with HIPEC is well-established, for PSM from other digestive malignancies, further high-quality studies are needed before recommending this approach outside clinical trials.

## Data Availability

Not applicable.

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
