# Peer review of "Surgical Options for Peritoneal Surface Metastases from Digestive Malignancies—A Comprehensive Review"

_medicina, 2023, doi:10.3390/medicina59020255_

Round 1

Reviewer 1 Report

In this paper, Mihai AE et al attempted to revise, in a narrative way, surgical options for peritoneal surface malignancies from digestive neoplasms. The review could be of interest, anyway there are several issues that need to be addressed as major concerns. 

English language needs immense revision throughout the entire paper. I would suggest to submit the paper to native English speaker or to institutional service before resubmission.

Abstract is well organized, however it should be rewritten in a more attractive manner. The abstract is the only part of the paper that readers see when they search through electronic databases such as PubMed. Finally, most readers will acknowledge, with a chuckle, that when they leaf through the hard copy of a journal, they look at only the titles of the contained papers.

Introduction section is well organized and cover near-all aspects of peritoneal metastases. It would of high interest, as well as completeness, to add some sentences as regard the HIPEC administered in a prophylactic setting. 

PCI – please use peritoneal cancer index instead of peritoneal carcinomatosis index throughout the entire paper. 

Surgical options for PSM from colorectal cancer -  the section appears well organized, even if it is too long to read with attention. Line 274: Results from Prophylochip and colopec trial have been published and I strongly invite Authors to update their review. Please use and add this extremely new and updated review in your paper: 10.3390/cancers15010165.

Surgical options for PSM from gastric carcinoma – Line 321-324: Authors reported multiple retrospective studies from European specialized centers. However, they lack to properly cite the multicenter study of Italian Peritoneal Surface Malignancies Oncoteam from Italian Society of Surgical Oncology (10.1245/s10434-021-10157-0 and 10.1245/s10434-021-10206-8). 

I would suggest Author to highlight, at the end of every paragraph (colon, stomach etc..) the main findings and conclusions giving readers the most important take home message. 

Author Response

Dear Reviewer

Thank you very much for your recommendations and suggestions which helped us to improve the academic quality of our manuscript!

  1. English language needs immense revision throughout the entire paper. I would suggest to submit the paper to native English speaker or to institutional service before resubmission.

Thank you very much for your suggestion! The entire manuscript has been revised by a native English speaker.

  1. Abstract is well organized, however it should be rewritten in a more attractive manner. The abstract is the only part of the paper that readers see when they search through electronic databases such as PubMed. Finally, most readers will acknowledge, with a chuckle, that when they leaf through the hard copy of a journal, they look at only the titles of the contained papers.

Thank you very much for this recommendation! We re-done the abstract.

  1. Introduction section is well organized and cover near-all aspects of peritoneal metastases. It would of high interest, as well as completeness, to add some sentences as regard the HIPEC administered in a prophylactic setting.

We added a paragraph regarding the use of HIPEC as a prophylactic approach in patients at high-risk for development of PSM.

“Furthermore, the paper reflects the most recent evidence regarding the prophylactic use of HIPEC in patients with colorectal or gastric carcinoma at high risk for developing PSM. We review the data critically, taking into account the limitations of the studies, and suggest future directions of research.”

  1. PCI – please use peritoneal cancer index instead of peritoneal carcinomatosis index throughout the entire paper.

We made the changes throughout the entire manuscript.

  1. Surgical options for PSM from colorectal cancer - the section appears well organized, even if it is too long to read with attention. Line 274: Results from Prophylochip and colopec trial have been published and I strongly invite Authors to update their review. Please use and add this extremely new and updated review in your paper: 10.3390/cancers15010165.

Thank you for these suggestions which improved the quality of the manuscript!

We divided this section in 7 sub-chapters.

We also discussed the results of the PROPHYLOCHIP and COLOPEC trials. We also took into account the information provided by the review that you have mentioned:

“The PROPHYLOCHIP-PRODIGE 15 trial [67] evaluated the impact of second-look surgery and HIPEC vs. follow-up on 3-yr DFS of patients with resected CRC and high-risk of developing PSM (perforated primary tumor/peritoneal or ovarian metastases radically resected concomitant with CRC). The authors did not find a significant difference in 3-yr DFS rates (44% vs. 53%, respectively; p value = 0.82). On the other hand, the COLOPEC trial [68] assessed the role of adjuvant HIPEC in preventing the occurrence of peritoneal metastases in patients with resected T4/perforated primary tumor, who received adjuvant systemic chemotherapy. There was no statistically significant difference in 18-months peritoneal DFS rates between patients treated with adjuvant systemic therapy only (76.2%) and those treated with adjuvant HIPEC and systemic therapy (80.9%; p value = 0.28). However, in both of these studies, as well as in the PRODIGE 7 trial, the HIPEC protocol consisted in administration of Oxaliplatin only for 30 minutes. Although these trials have generated skepticism towards the usefulness of HIPEC, these results could be challenged by ongoing/future trials evaluating different protocols of HIPEC. Until new HIPEC protocols will be tested in well-designed comparative trials, this procedure should not be considered an ineffective method. [69]”

[67] D. Goéré, O. Glehen, F. Quenet, J. M. Guilloit, J. M. Bereder, G. Lorimier, E. Thibaudeau, L. Ghouti, A. Pinto, J. J. Tuech, R. Kianmanesh, M. Carretier, F. Marchal, C. Arvieux, C. Brigand, P. Meeus, P. Rat, S. Durand-Fontanier, P. Mariani, Z. Lakkis, V. Loi, N. Pirro, C. Sabbagh, M. Texier, D. Elias, "Second-look surgery plus hyperthermic intraperitoneal chemotherapy versus surveillance in patients at high risk of developing colorectal peritoneal metastases (PROPHYLOCHIP-PRODIGE 15): a randomised, phase 3 study.," Lancet Oncol, vol. 21, no. 9, pp. 1147-1154, Sep. 2020.

[68] C. E. L. Klaver, D. .D. Wisselink, C. J. A. Punt, P. Snaebjornsson, J. Crezee, A. G. J. Aalbers, A. Brandt, A. J. A. Bremers,J. W. A. Burger, H. F. J. Fabry, F. Ferenschild, S. Festen, W. M. U. van Grevenstein, P. H. J. Hemmer, I. H. J. T. de Hingh, N. F. M. Kok, G. D. Musters, L. Schoonderwoerd, J. B. Tuynman, A. W. H. van de Ven, H. L. van Westreenen, M. J. Wiezer, D. D. E. Zimmerman, A. A. van Zweeden, M. G. W. Dijkgraaf, P. J. Tanis, "Adjuvant hyperthermic intraperitoneal chemotherapy in patients with locally advanced colon cancer (COLOPEC): a multicentre, open-label, randomised trial.," Lancet Gastroenterol Hepatol, vol. 4, no. 10, pp. 761-770, Oct. 2019.

[69] A. Sommariva, M. Tonello, F. Coccolini, G. De Manzoni, P. Delrio, E. Pizzolato, R. Gelmini, F. Serra, E. Rreka, E. M. Pasqual, L. Marano, D. Biacchi, F. Carboni, S. Kusamura, P. Sammartino, "Colorectal Cancer with Peritoneal Metastases: The Impact of the Results of PROPHYLOCHIP, COLOPEC, and PRODIGE 7 Trials on Peritoneal Disease Management.," Cancers (Basel), vol. 15, no.1, pp. 165, Dec. 2022.

  1. Surgical options for PSM from gastric carcinoma – Line 321-324: Authors reported multiple retrospective studies from European specialized centers. However, they lack to properly cite the multicenter study of Italian Peritoneal Surface Malignancies Oncoteam from Italian Society of Surgical Oncology (10.1245/s10434-021-10157-0 and 10.1245/s10434-021-10206-8).

Thank you for your recommendation! We up-dated the review with the results of this trial (that was also cited):

“A more recent multicentric study from Italy, which includes 91 patients with gastric carcinoma and synchronous PSM, reports median OS rates higher than 40 months in patients with a PCI ≤6, as well as in those who underwent complete cytoreduction [82]. Thus, the median OS after CC-0 is significantly higher compared to the OS of patients with incomplete resection (40.7 vs. 10.7 months, respectively; p value = 0.003). Moreover, in patients with a PCI > 6 the median OS was significantly lower than in patients with a PCI ≤6 (13.4 vs. 44.3 months, respectively; p value = 0.005) and the mortality was almost double [82]” …

… “A recent multicenter study of “Italian Peritoneal Surface Malignancies Oncoteam—S.I.C.O.” proved the beneficial effect of neoadjuvant chemotherapy on long-term outcomes of patients eligible for CRS and HIPEC [-0]. The same group highlighted a significant negative prognostic effect determined by positive peritoneal cytology [82] [88].”

[82] L. Marano, D. Marrelli, P. Sammartino, D. Biacchi, L. Graziosi, E. Marino, F. Coccolini, P. Fugazzola, M. Valle, O. Federici, D. Baratti, M. Deraco, A. Di Giorgio, A. Macrì, E. M. Pasqual, M. Framarini, M. Vaira, F. Roviello, "Cytoreductive Surgery and Hyperthermic Intraperitoneal Chemotherapy for Gastric Cancer with Synchronous Peritoneal Metastases: Multicenter Study of 'Italian Peritoneal Surface Malignancies Oncoteam-S.I.C.O.," Ann Surg Oncol., vol. 28, no. 13, pp. 9060-9070, Dec. 2021.

[88] L. Marano, D. Marrelli, F. Roviello, "ASO Author Reflections: Gastric Cancer with Synchronous Peritoneal Disease-A Clinically Meaningful Survival After CRS and HIPEC in Selected Patients from Italian Peritoneal Surface Malignancies Oncoteam Network.," Ann Surg Oncol., vol. 28, no. 13, pp. 9071-9072, Dec. 2021.

  1. I would suggest Author to highlight, at the end of every paragraph (colon, stomach etc..) the main findings and conclusions giving readers the most important take home message.

We added at the end of each chapter a “take-home message” which reflects the main findings and conclusions of the chapter.

Sincerely yours,

Sorin Tiberiu Alexandrescu, MD, PhD

Senior lecturer at Department of Surgery, Faculty of Medicine, Carol Davila University of Medicine and Pharmacy, 050474 Bucharest, Romania

Head of the 1st Department of General Surgery, Fundeni Clinical Institute, 022328 Bucharest, Romania

January 25, 2023

Reviewer 2 Report

General points:

This is a comprehensive review article focused on surgical options for peritoneal surface metastases (PSM) from various digestive malignancies. The malignancies included colorectal cancer, gastric carcinoma, pseudomyxoma peritonei, pancreatic adenocarcinoma, biliary tract carcinoma, gastrointestinal stromal tumors, gastroenteropancreatic neuroendocrine tumors, and small bowel adenocarcinoma. The main concept of surgical option is a cytoreductive surgery, and more enhanced treatment including HIPEC and PIPAV have developed. By reading this manuscript, clinicians and surgeons can broadly update their knowledge regarding this topic.

Please consider the following points for the next manuscript processing.

Specific points:

1.      Although this manuscript is a narrative review without systematic approach, the literature search method can be shown (e.g., year, database, study design).

2.      Regarding colorectal cancer (other malignancies are expressed as “carcinoma”) and gastric carcinoma, the contents extended over 2-4 pages. These may not be easy to read without subheadings. The authors can divide the sections using subheadings (e.g., etiology, incidence, treatment regimens, predictors of response, prognosis…) to draw and maintain readers’ attention. Tables of available data may be hepful. 

3.      Page 8, Line367-8: “3. a HIPEC as a prophylactic… developing PSM”, it seems one of the subheadings but this is the only one throughout this manuscript.

4.      The conclusion section in this manuscript is a summary of the most relevant findings of each malignancy. The conclusion section would be more applicable if it shows an overview of the subject as well as arguments, limitations and future directions.

Author Response

Dear Reviewer

Thank you very much for your recommendations and suggestions which helped us to improve the academic quality of our manuscript!

  1. Although this manuscript is a narrative review without systematic approach, the literature search method can be shown (e.g., year, database, study design).

Thank you very much for your valuable recommendation! We added a new chapter to report the search method that we used:

“2. Paper selection

                We searched the PubMed database, using the following terms: (((((((((((((peritoneal surface metastasis[Text Word]) OR (carcinomatosis[Text Word])) AND (colorectal cancer[Text Word])) OR (gastric carcinoma[Text Word])) OR (digestive malignancies[Text Word])) OR (biliary tract carcinoma[Text Word])) OR (pancreatic carcinoma[Text Word])) OR (gastrointestinal stromal tumors[Text Word])) OR (neuroendocrine tumors[Text Word])) OR (small bowel carcinoma[Text Word])) AND (cytoreductive surgery[Text Word])) OR (HIPEC[Text Word])) NOT (ovarian cancer[Text Word])) NOT (mesothelioma[Text Word]). The filters applied were: Clinical Trial, Meta-Analysis, Randomized Controlled Trial, Review, Systematic Review, from 2001/1/1 to 2022/6/1. The search generated 538 results. The abstracts of these results were evaluated by two authors (M.A.E. and G.P.), the relevant papers were extracted independently and their full-text versions were assessed. Consensus for the relevance of a study was carried out by the third author (S.T.A.). We also evaluated the references of the relevant papers were evaluated in order to identify additional articles that were not found during the initial search. Due to the heterogeneity of the studies, we report the results as a narrative review.”

  1. Regarding colorectal cancer (other malignancies are expressed as “carcinoma”) and gastric carcinoma, the contents extended over 2-4 pages. These may not be easy to read without subheadings. The authors can divide the sections using subheadings (e.g., etiology, incidence, treatment regimens, predictors of response, prognosis…) to draw and maintain readers’ attention. Tables of available data may be hepful.

We made the correction and replaced colorectal cancer with the term “colorectal carcinoma (CRC)”. We divided these sections in sub-chapters.

  1. Page 8, Line367-8: “3. a HIPEC as a prophylactic… developing PSM”, it seems one of the subheadings but this is the only one throughout this manuscript.

We divided these sections in sub-chapters.

  1. The conclusion section in this manuscript is a summary of the most relevant findings of each malignancy. The conclusion section would be more applicable if it shows an overview of the subject as well as arguments, limitations and future directions.

We re-done the Conclusion section, guided by your recommendation:

“The aggressive surgical approach of PSM from digestive malignancies, consisting in CRS with or without HIPEC, has gained wider acceptance during the last decade, especially in patients with CRC or gastric carcinoma. This is the consequence of the evidence offered by high-quality randomized clinical trials and meta-analysis which revealed that CRS is the cornerstone therapy in patients with PSM from CRC, although the oxaliplatin-based HIPEC regimen failed to further improve the survival of these patients. Supplementary well-designed randomized trials testing new HIPEC regimens are needed before refuting this therapy. Similarly, in patients with PSM from gastric carcinoma, future randomized controlled trials are needed to confirm the favorable outcomes achieved by CRS with HIPEC in large retrospective studies and meta-analysis. While in PMP the role of CRS with HIPEC is well established, for PSM from other digestive malignancies further high-quality studies are needed, before recommending this approach outside clinical trials.”

Sincerely yours,

Sorin Tiberiu Alexandrescu, MD, PhD

Senior lecturer at Department of Surgery, Faculty of Medicine, Carol Davila University of Medicine and Pharmacy, 050474 Bucharest, Romania

Head of the 1st Department of General Surgery, Fundeni Clinical Institute, 022328 Bucharest, Romania

January 25, 2023

Reviewer 3 Report

In this article, Eftimie et al. provided a narrative review of the various available surgical options for peritoneal metastates from digestive tumors. The article is well written and properly organaized. Moreover it includes all the recent evidence from trials and pooled analyses regarding the safety and efficacy of CRS and intraperitoneal chemotherapy in these malignancies. Despite the overall quality of the article, my considerations include issues regarding the scientific novelty of the study; the authors should comment on this issue and support their effort compared to other similar published reviews

Author Response

Dear Reviewer

Thank you very much for your recommendation which was very useful!

In this article, Eftimie et al. provided a narrative review of the various available surgical options for peritoneal metastases from digestive tumors. The article is well written and properly organaized. Moreover, it includes all the recent evidence from trials and pooled analyses regarding the safety and efficacy of CRS and intraperitoneal chemotherapy in these malignancies. Despite the overall quality of the article, my considerations include issues regarding the scientific novelty of the study; the authors should comment on this issue and support their effort compared to other similar published reviews.

Although there are similar reviews published on this topic, a lot of patients who are candidates for such an aggressive approach lose the chance for such treatment because many general practitioners, gastroenterologists and even oncologists or surgeons do not know the recent results of this therapy. Thus, we consider that publication of such a review in a journal included in the category “Medicine, General & Internal” could be useful, aiming to advance the field by informing current practice and by prompting clinicians to act and broaden the use of an aggressive surgical approach in patients with PSM from digestive carcinomas. We stated this in the Introduction section of the manuscript:

“Although the issue of CRS +/- HIPEC for PSM from specific malignancies was addressed in other recent papers, there is a paucity of reviews which present together the latest evidence regarding the surgical options for all digestive carcinomas with peritoneal metastases. This paper aims to advance the field by informing current practice and by prompting clinicians to act and broaden the use of an aggressive surgical approach in patients with PSM from digestive carcinomas. Given the current evidence, concerted efforts should be made by general practitioners, gastroenterologists, oncologists and surgeons to promote the CRS with or without HIPEC in order to prolong the life-expectancy of these patients.”

Sincerely yours,

Sorin Tiberiu Alexandrescu, MD, PhD

Senior lecturer at Department of Surgery, Faculty of Medicine, Carol Davila University of Medicine and Pharmacy, 050474 Bucharest, Romania

Head of the 1st Department of General Surgery, Fundeni Clinical Institute, 022328 Bucharest, Romania

January 25, 2023

Round 2

Reviewer 1 Report

The paper has been improved according to the reviewer suggestions.